# Fashioning the Circular Economy with Disruptive Marketing Tactics Mimicking Fast Fashion's Exploitation of Social Capital: A Case Study Exploring the Innovative Fashion Rental Business Model "Wardrobe"

**Elaine L. Ritch [1],* and Noreen Siddiqui [2]**

[1] Glasgow School for Busienss and Society, Glasgow Caledonian University, Glasgow G4 0BA, UK
[2] Adam Smith Business School, University of Glasgow, Glasgow G12 8QQ, UK; noreen.siddiqui@glasgow.ac.uk
* Correspondence: elaine.ritch@gcu.ac.uk

**Abstract:** With the threat of the climate emergency intensifying and limited time left to reduce irreversible consequences, the need to consider how natural resources are excavated and managed from cradle to grave intensifies. This positions the circular economy (CE) as being highly relevant, particularly for the fashion industry, which is criticised for encouraging continued frequent and impulsive consumption of inexpensive garments with limited longevity. Advancing the circular fashion economy (CFE) has received little attention. Limited research to date has found that consumers have not been socialised to consider fashion acquisition as a collaborative or sharing activity, revealing an established attitude–behaviour gap that prohibits the advancement of the sustainable-fashion agenda. Primarily, fashion is imbued with social and emotional capital, as experienced with the dominant social paradigm (DSP) of fast fashion. This paper argues that similar tactics can be adopted for sustainable fashion practices through the CFE by exemplifying a case study of a fashion-renting platform, "Wardrobe," that enables consumers to rent fashion owned by influencers and celebrities. In doing so, the paper makes four contributions to the knowledge: Firstly, in developing a conceptual framework from research examining fashion, sustainable fashion, and the CFE, the paper illuminates how fashion marketing emphasises social and celebrity capital to appeal to consumer emotions, encouraging frequent impulsive consumption, and how this can be transferred to the CFE. Secondly, the DSP is contextualised alongside the theory of disruptive innovation to understand how social norms of fashion consumption can be disrupted. Thirdly, although there is an emerging literature stream examining the CE and CFE, this focuses more on consumer practice and behaviours, and little attention has been paid to how the CFE can be marketed to engage with consumers. Fourthly, this paper illuminates how similar marketing tactics used by fast fashion can be exploited to advance the CFE.

**Keywords:** circular economy; fashion; disruptive innovation; social capital; renting; celebrity; influencers

## 1. Introduction

The greatest challenge facing the world at present is the threat of the climate emergency disrupting the stability of the planet [1]. Tackling this issue requires transformational change underpinned by sustainability principles and prioritised by global governance, business, and citizen practice [1]. The circular economy (CE) adopts sustainability practices by maximising finite resources and increasing product lifetimes as well as decreasing waste sent to landfills. However, as there has been little research into the implementation of the CE, especially from a fashion context [2], it follows that there has also been little acknowledgement of how the CE can be marketed to appeal to consumers. The research that has been carried out on the circular fashion economy (CFE) has identified numerous barriers, and it appears that sustainable fashion is still a niche market [2,3], dominated

by the fast fashion business model [4]. To date, sustainable fashion has sought to appeal to consumers' moral and ethical values, underpinned by their concern for the climate emergency [5]. In this article, we explore how the CFE can compete with fast fashion by adopting a similar marketing strategy that offers consumers viable alternative option to acquire fashion.

The fast fashion industry is inherently unsustainable, and efforts to address sustainability are considered inauthentic in tackling the core issues [6]. Fast fashion marketing tactics encourage frequent impulsive consumption of inexpensive garments [7,8], of which production is reliant upon scarce resources and exploitative practices (of people and the planet) [6]. Low pricing increases notions of disposability, and consumers have been socialised to acquire and dispose of fast fashion frequently [9] in what have become established social norms of fashion practices [10]. This once innovative and competitive fast fashion business model is now the dominant social paradigm (DSP) of fashion production and consumption, and both the industry and consumers are locked into this system [5]. Fast fashion sustainability is a superficial response, passing responsibility to consumers to donate unwanted garments to charity rather than take responsibility to integrate sustainability into production and support disposal routes [11,12]. Yet, the volume of clothes donated to charity is significantly greater than consumers buying second-hand fashion [3,13]. As discussed below, the literature presents clearly defined barriers preventing participation in alternative fashion practices [14,15]. Advances in the CFE would make the most of scarce resources in circulation and reduce exploitative practices of people and the planet [2,3]; however, there is little understanding of how this might be actualised within fashion practices, especially when competing with the DSP of fast fashion marketing and retailing and the inclusion of social and aesthetic capital, which underpins involvement in fashion.

This paper proposes that, to advance the CFE, similar tactics that encourage the social and aesthetic capital of fast fashion consumption are necessary for competitiveness; there are lessons to be gleamed from the way in which the fast fashion industry operates that can be utilised to disrupt the DSP. The paper argues that, just as fast fashion disrupted the fashion industry, resulting in a race to the bottom [4], innovation can break through this DSP to create new fashion practices that offer alternative distribution channels that will appeal to consumers' sustainable values. Previous research has outlined an existential vacuum in which the fashion industry and fashion consumers stagnate, aware of the fashion industry's contribution to the climate emergency but compelled to continue within the DSP for fear of losing market share and not conforming to social norms [5,14,15]. Research has examined fashion engagement, demographics, values, life stage, and knowledge of the issues (see [13] for an overview) but has found no clarity in what encourages sustainable fashion practices, as recognised in the well-established attitude–behaviour gap [16]. Although change in societal, institutional, and business behaviours are necessary, we believe that learning and replicating how fast fashion marketing has captured consumers' attention to entice rapid fast fashion consumption can innovate the CFE to disrupt the incumbency of the DSP and provide a resolution to this reported attitude–behaviour gap. Therefore, this paper will present an established CFE business, "Wardrobe, as a case study. Wardrobe is an online a peer-to-peer fashion-sharing platform operating in the USA that offers consumers access to luxury brands as well as celebrity wardrobes. We examine Wardrobe's business model through a conceptual framework that was constructed from reviewing the fashion and marketing literature along with the theoretical framework of disruptive innovation developed by Christensen [17]. Disruptive innovation has been successful in determining shifts in the marketplace that are supported by technological advancement and provides an opportunity to understand market innovations that disrupt current ways of business operations. Market expansion is often sought by moving into new geographical areas, yet there can be new forms of business operations that can support growth, and this has been evident in disruptive business models in the sharing economy, such as Uber and Airbnb. Goffin, Åhlström, Bianchi, and Richtnér [18] assert that case study research can help to explain new concepts of business model innovation. The paper will first review extant

literature to construct a framework of fast fashion marketing concepts that can underpin a CFE and then outline the theory of disruptive innovation and its application within the fashion sector. This will be followed by introducing Wardrobe and demonstrating how this CFE utilises similar tactics and terminology as those f fast fashion marketing to redirect consumers' fashion practices. The paper will conclude with implications for theory, fashion marketing, and retailing.

## 2. Conceptual Framework

Before presenting the case study of Wardrobe, it is important to establish what needs to be disrupted by the CFE by examining how the DSP of the fast fashion industry operates and engages with consumers. Firstly, the paper will consider the social context, and having reviewed the fashion and marketing literature, it is concluded that it is Generation Z who mainly purchase fast fashion while also expressing heightened concern for the climate crisis [14,19]. This dichotomy [20] underpins the conceptual framework by illuminating the social constructs in which fast fashion operates; however, it must also be recognised that this social context includes a communicative interactive process that signals subliminal messages to others—representing self-identity that is both presented and seen socially [5,15]. Secondly, to encompass emotive contexts, this paper considers the role of online presentation on social media platforms that present self-identity and fashion through extended geographical social networks. Social media offers opportunities for everyone to have a platform and a voice but also enables increased competition and insecurities [21]. Generation Z have vast quantities of information available to them, as well as increased social pressure [22]. The literature suggests that social and aesthetic capital are more powerful constructs than values relating to sustainability concerns [15], and therefore appealing to ethical value has so far been unsuccessful in challenging the DSP of fast fashion [23]. Our conceptual framework comprises three elements, as presented in Figure 1, which will be examined in the sections below. The conceptual framework emerged from extant literature and focuses on understanding the context and concepts that require disruption to enable a new understanding of how to position the CFE as appealing to consumers.

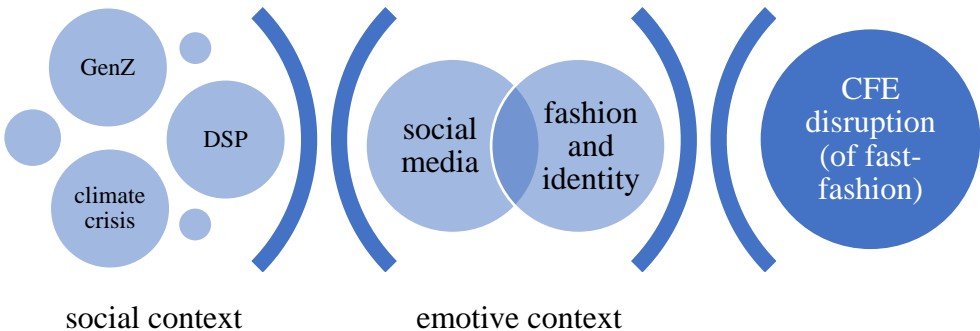

**Figure 1.** Conceptual framework: understanding the social and emotive contexts of fast fashion.

### 2.1. Social Context

As represented in Figure 1 above, the social context represents the external influences that influence DSPs. As noted above, research examining demographics and psychographics are inconclusive on the variables that make ethical consumption more likely. Although interest in sustainable fashion has increased [14,19,20,24], it is insignificant when compared to the DSP of the fast fashion industry [3,4]. As the main consumers of fast fashion, Generation Z are also the focus of fast fashion marketing and more responsive to the tactics [13]. Generation Z, born between 1995 and the early 2010s, are digital natives and are prolific social media users [8]. However, Generation Z are also said to be more interested in sustainability as they inherit a world impacted by the climate emergency and seek justice and equality both socially and environmentally [19,20]. This contradiction of driving fast fashion sales while simultaneously experiencing anxiety about the climate

emergency manifests as cognitive dissonance [14] and is bound within the DSP of social norms and expectations that provoke emotional experiences. The dichotomy reflects the lure of social capital, which portrays the self as belonging in wider society and is indicative of social relationships [25], constructs that impact self-esteem and confidence [26] and induce emotional responses. This social context underpins the emotive context, where the cycle of consumption is encouraged and guilt is experienced [13].

### 2.2. Emotive Context

Understanding the relationship between fashion and identity, status, self-esteem, and confidence is fundamental to consumer engagement with fashion, and this is what sets fashion aside from other consumption contexts [5]. Emotions are triggered by fashion marketing [13] and social interactions, which appraise self-presentation as a process that is not always self-aware and yet construct perceptions of an individual's value in wider society [27]. Zhao [27] likens this to Cooley's [28] looking glass theory, where self-perception is created to manage others' views and is peer endorsed. This links to theories around conspicuous consumption [29], where it is recognised that commodities are used as props in creating a narrative that communicates self-identity and related values, signalling social cohesion and relationships [28]. Social media platforms offer consumers a place to create their identity, using imagery, commentary, and tags to people, places, and brands [8] and thus constructing a sense of self-worth and belonging in society that accumulates in communicating social capital.

### 2.3. Social Capital Representation on Social Media

Although elements of communicating social capital have underpinned self-presentation and identify formation for over a century (i.e., [29,30]), arguably, this has intensified through digital technology and social media [27,31]. The emergence of social media influencers as "taste makers" [4] is the evolution of celebrity capital [21] that has been at the fore of fashion movements—historically, from wealth, status, and royalty [32] to attributes relevant in various genres, i.e., music, sport, and fashion, for which celebrities are recognised [33]. The "power of celebrity in driving economic value" [34] has been noted by marketing, and fashion has both created celebrities (designers and models) and utilised celebrities for social and economic capital [21]. Trickling down from haute couture, fast fashion retailers began to collaborate with well-known celebrities to promote bespoke collections [35], and this evolved to co-creating collections with well-known societal personas (influencers/celebrities) that are marketed on social media [4] to the benefit of the brand and celebrity capital [21]. McFarlane, Hamilton, and Hewer [36] describe this as a form of micro-entrepreneurship that can market to a mass audience to illustrate their "passion for fashion." Influencer culture depicts an enviable lifestyle [37], where fashion plays a central role in forming social capital [36], and this encourages rapid trend change, which has led to the shame of re-wearing an outfit for Generation Z [13,14,20]. Through this lens, economic capital is generated by combining brand/celebrity capital within the aesthetic economy [38]. This powerful combination influences engagement with fashion and endorses expectations of the fast fashion DSP that encourages impulsive frequent consumption and, ultimately, disposal. Underpinning this is the emotive entanglement of presenting the self to society.

### 2.4. Emotional Capital of Fashion Engagement

Extant research has found that consumers are not prepared to sacrifice their identify for ambiguous claims that cannot be substantiated [5,23]. Generation Z often feel locked into fashion consumption due to marketing tactics positioning a sense of urgency along with offering hedonic experiences [9,13,39]. Low pricing encourages frequent impulsive consumption with minimal risk and is often stimulated by marketing tactics: flash time-bound offers, free delivery, slice-it interest-free credit [7,39]. The price of fast fashion has been decreasing over the last few decades, instigating notions of disposability that underpin the urgency to wear something once and then move onto new fashion [22].

Further stimulation of the frequent impulsive fashion consumption process is found in the hedonic emotive response that fashion acquisition provides. For example, Ritch et al. [13] found that consumers experience excitement when waiting for their purchase to arrive, and this unwrapping is often captured on social media for entertainment [24]. Engaging on social media is often a passive act, prompted by marketing that draws the consumer into social media, where consumption is just a few clicks away [8]. Coupled with the ability to return garments without cost and often not paying for the order until a few weeks later minimise the risk of fast fashion consumption [13]. Consequently, social media provides a blurring of entertainment and commerciality, and the CFE would benefit from recognising this.

By playing on social capital and the relationships between people and their place within society [25], the implications of emotions are both intrinsic and extrinsic in that fashion is marketed as being consumer centric, advocating the construction of the self as a communicative signal to wider society [5]. This is evident in fashion marketing, such as discount "treats" on pay day that one "deserves" for working hard [5]. Marketing further plays with emotions and feelings by situating fashion consumption as signalling success that represents "the good life" [28] often embodied by celebrities and, more recently, social media influencers [5]. Fashion hauls—videos in which influencers show a large volume of garments as a treat (see [24])—feed into marketing messages that obtaining new fashion is desirable and deserved, constructing the social norms and expectations of behaviour that formulate the DSP. If the DSP is to be disrupted, then social media is pivotal, as it fuels fast fashion consumption.

Brooks et al. [21] differentiate celebrity from influencer culture as being less about fame and more about being "attention-worthy," which converts into being "profit-worthy" (p. 537), whereby content drives the attention economy [33]. Authenticity is central to fostering a strong relationship between brands and consumers, and influencers often aid authenticity through deepening sincerity and trust to form attachment. Cumulatively, it is the social and the emotive context, displayed in Figure 2, that have hindered the development of the CFE. Rather than appealing to consumer values that are underpinned by concern for the climate emergency, there is the potential to devise a CFE business model that encapsulates the components of social and emotional capital. As will be discussed next, although there have been advances in the CFE, they is insignificant in combating the volume of fast fashion sold and disposed of within the DSP of social norms.

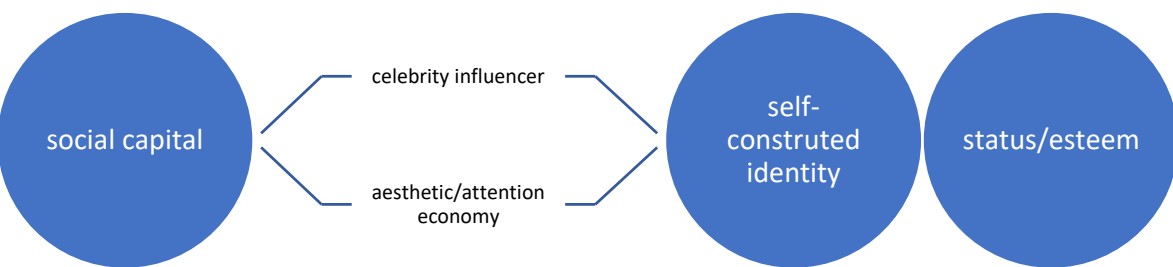

**Figure 2.** Components of fast fashion-induced social capital.

### 2.5. Circular Fashion Economy

Viewed from a lifecycle perspective—from the cradle to the grave—there have been calls for the fashion industry to implement a "self-contained closed loop system" of production to consider ways in which the lifespan of scarce resources can be extended [40] (p. 610). Although there have been advances in sustainable fashion production, the climate crisis cannot be solved with more consumption [41]. To be environmentally responsive, the focus should be on the "efficiency and sufficiency" of scarce resources [42] (p. 189) to prevent premature disposal to landfills [43]. This closed loop represents the sharing economy, where collaborative consumption ensures that more than one person benefits from the garment's lifespan [3,44]. Although this can include sharing and swapping in formal

monetary systems or as goodwill, such as redistribution markets that include peer-to-peer and commercial networks, this paper will focus on a peer-to-peer commercial model of renting as fashion sharing [45]. Renting is defined as "one party offers an item to another party for a fixed period in exchange for a fixed amount of money and in which there is no change of ownership" [46] (p. 90). Although traditionally, renting has been more common for luxury or special occasion garments, such as a prom or wedding outfit [3,47], participation has been made easier due to digitalised platforms that enable easier access and more variety [3,43,48].

The CFE model of renting has the potential to disrupt the DSP of fast fashion, and growth in business models embracing renting is growing and is expected to increase further [2,3]. Jain et al. [2] carried out a systemic literature review on the mainstreaming of fashion rental to better understand how consumers engage with the practice. They identified four aspects of interest. Firstly, their review covered only 41 academic papers, suggesting that there has been little attention paid to the CFE model of renting. Secondly, they found significant recent growth in research on consumer perceptions of renting fashion in the last few years, demonstrating the current attention being paid to this business model as responsive to the sustainable fashion agenda [2,3]. Thirdly, many of the papers adopted a conceptual approach, asking consumers' opinions of renting from an abstract perspective, as most did not have experience with this fashion practice. However, neither Jain et al. [2] nor Arrigo [3], who examined digital fashion-renting platforms in Italy, identified a business model that harnessed the aesthetic and social capital of influencers/celebrities. They also noted that most of the research focused on physical renting rather than online platforms. Finally, most studies have focused on business-to-consumer (B2C) renting platforms, with very limited attention paid to peer-to-peer (P2P) renting platforms [2,3].

Research has identified the barriers as restricting advancement of the model; for example, entrenchment within social norms of consumption, made possible through low pricing and easy access, has led to preferences for ownership [42], which also offers minimised risk, both financial and convenience [13,22]. The accumulation of fashion is illustrative of social capital and "the good life," as represented in fashion haul videos where quantity is valued over quality; again, this is made possible because fast fashion is less expensive to purchase that fashion is to rent [43]. Consumers have also expressed fear of damaging rented clothing, especially from the luxury sector [2,49], and are deterred from the limited options available for renting, such as sizing and fit, along with lack of awareness and uncertainty of the process of renting [42,50]. However, research is scarce, especially as many consumers are unfamiliar with the concept, part from renting special occasion outfits, and this has led to the CFE being an underdeveloped business model [50]. Jain et al. [2] noted that many of the research participants in their systematic review were reflecting on the notion of renting rather than reporting on actual experience, as was found in the research carried out by Westerberg and Martinez [50] on young German consumers, which limits the expansion of fashion rental. Nevertheless, there was acknowledgment of the hedonic values that could be obtained from renting fashion, especially given the variety of options that could provide identity exploration without the expense and risk of consumption [2,3]. Jain et al. [2] and Arrigo [3] also reported that it offers a route to luxury fashion that is unaffordable to purchase, and as such responds well to egotistic value.

## 3. Theoretical Framework: Disruptive Innovation

Christensen's [17] seminal theory of disruptive innovation (DI) has been much cited over the last few decades, as well as being challenged on theoretical and practical elements (see [51]). Nevertheless, the theory does enable a closer examination on how to disrupt DSPs through the role of innovation and carve out new value propositions to support the CFE. As presented in Figure 3 below, the theory of DI sits amongst other types of innovation theory that are utilised to solve problems; what makes DI more relevant for this paper is how it can advance technological and marketplace shifts [52]. This is evident in how the fast fashion industry became possible through globalisation and technological advances,

where low pricing was facilitated by cutting corners on good practices that protected the environment and workers. More recently, technology and the marketplace has shifted again through instant global communications accessible on smartphones [3] and offers a platform to share the growing concern for the climate emergency.

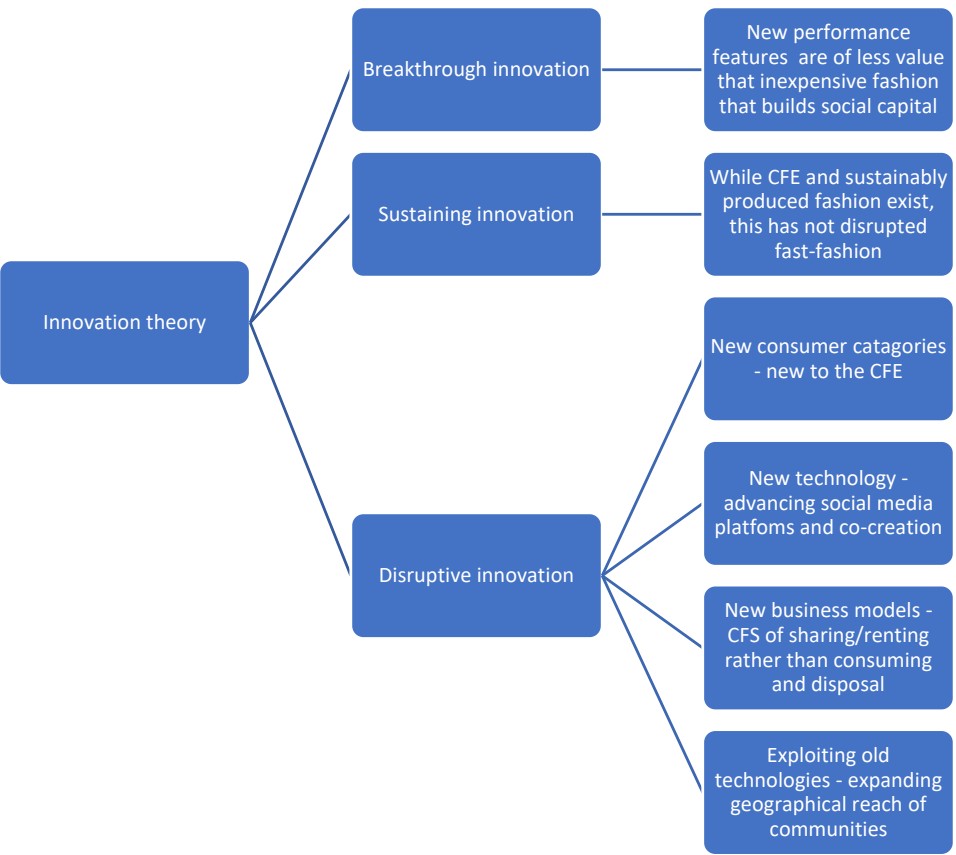

**Figure 3.** Determining which innovation theory advances the CFE.

Satell [52] advises that when a problem is well defined (i.e., fast fashion produced through exploiting people and the planet and marketed to encourage frequent consumption and disposal), breakthrough innovation can help provide a solution. Breakthrough innovations introduce a new set of performance features to transform existing markets. Yet, sustainably produced fashion remains niche due to barriers of fashionability, price, and accessibility, as established above, and has not drawn consumers from the DSP despite their concern for the climate emergency. Therefore, new sustainable features have not resolved the problem. Conversely, when it is the domain that is well defined, disruptive innovation is more suitable [52]. Although the problems of fast fashion regarding the climate crisis are well defined and increasingly filtering into social discourse, it is not enough to educate consumers on the impact and expect them to make sustainable fashion choices when it requires a sacrifice in price and style [5,14,22]. Further, the new set of performance features of sustainable fashion do not address the social and emotive capital imbued in fashion engagement. Although some fast fashion retailers have introduced facets of sustainable fashion, such as one sustainable collection that is dominated by continued production of fast fashion, it is marketed on the same principles as fast fashion: cheap, fast, and of poor quality, and this superficial response is considered inauthentic [5,11,12]. Therefore, as addressing the problem definition has been unsuccessful, it may be better to consider the domain—the marketplace—and disrupt the DSP by using similar social capital marketing tactics.

Christensen [17] proposes two types of innovation: sustaining, either evolutionary or revolutionary, and disruptive. Sustaining innovations do not impact existing markets but offer improvements to products and services that evolve within markets or transform markets, often at the high-end and luxury sectors, without disrupting existing market players [51]. This is illustrated in sustainable fashion, which occurs more authentically in higher-priced brands. Although it could be assumed that the fast fashion business model was disruptive, in driving down the price of fashion to be more affordable, convenient, and more responsive to trend changes, it does not align with Christensen's [17] framework, as will be discussed below. Rather, sustaining innovations are prevalent in the fast fashion industry, competing in terms of price in a homogeneous and crowded market. Similarly, sustainable fashion has not disrupted the DSP of the fast fashion business model; rather, sustainable fashion is perceived as more expensive, not fashionable, and less convenient to access and aligns with high-end encroachment theory, as outlined by Schmidt and Druehl [51]. These established barriers support the cycle of unsustainability that the fashion industry and fashion consumers are locked into [5,9]. Therefore, to challenge the DSP, disruption is required, and the next section will examine Christensen's [17] framework to consider how to provide similar emotions and experiences to challenge fast fashion.

The theory of disruptive innovation creates a new market or enters at the bottom of an existing market, and Christensen et al. [53] stipulate four elements that underpin disruption. The first is new consumer categories that can be attracted by new technology and business models. Christensen et al. [53] provide the example of computers moving into consumer homes, whereas previously they were only owned by businesses, opening up a new consumer market. Mitzkus [54] considers Ford to have disrupted the automobile industry by introducing a moving assembly line that reduced the price of production, enabling automobiles to become more affordable and enter the mass market. Although it can be assumed that fast fashion was disruptive by reducing the price and ensuring that consumption is more accessible, the model has cannibalised production to encourage frequent impulsive consumption that sustains the industry rather than creating new consumers [4]. Historically, when fashion was not as affordable, consumers bought less, shared more, and repaired damaged garments [15,23]; fast fashion has simply accelerated the cradle-to-grave process. Disruption of new consumer categories will come from the CEF by designing a new business model of acquiring and practicing fashion that offers a viable sustainable and affordable route to acquire fashion through an innovative business model.

The second is new technologies that use technology to better meet consumer needs. Christensen et al. [53] provide the example of Netflix moving from lending physical DVDs to consumers through postal systems to the instantaneous act of streaming films though a subscription service. This convenience and expansion of product availability offered a USP that provided a competitive advantage [55]. Similarly, Mitzkus [54] notes the Apple iPod as a technological development that, although smaller, enabled consumers to access a wider variety of portable music than the personal portable DVD and cassette players. Although the iPod was initially expensive [51], the price points were reduced within a variety of models. New technologies also played a role in lowering the price of fast fashion and quickening the production process to respond to rapid trend changes. However, this represents sustained innovation that accelerated fast fashion consumption, exacerbating the exploitation of the environment and workers rather than using technology in a new way to enhance the consumer experience. Although fast fashion retailers harness technology to augment the consumer experience by minimising risk and effort for consumers and offering personalised marketing [13], the aim here is to encourage more consumption, which does not support the sustainability agenda in addressing the issues that are of concern to Generation Z. Arguably, this capitalistic model benefits fast fashion business owners and shareholders more than consumers [56], who profit from the insecurities of social capital, and this expansion of social inequalities and lack of diversity does not sit well with Generation Z [19].

New business models are the third element of Christensen's theory [17], and examples include Alibaba, Uber, Airbnb, and Facebook as business models that have eliminated the need for physical assets; rather, they act as a conduit between producers, retailers, and consumers, and this is supported by digital technology [3]. The Alibaba group (the Chinese equivalent of Amazon), the worlds' largest retailer, does not own any goods; Uber, the worlds' biggest taxi group, does not own any vehicles [54]; Airbnb, the worlds' biggest provider of accommodation, does not own any real estate; and Facebook, the worlds' biggest media company, does not create any content [57]. There are lessons for the fashion industry from these examples that support the CE by maximising the utility of garments' lifespans and making the most of resources before they end up in landfills. In contrast, the fast fashion model, hailed as new in streamlining production efficiencies and increasing the competitiveness of low pricing, has reduced consumer thresholds of how much they were willing to pay [23]. Fast fashion is reliant on frequent impulsive consumption to rotate their inventory, as profit comes from cutting production costs and generating sales. Taking inspiration from the four examples provided above in which disruptive business models do not accumulate physical assets, unlike fast fashion, "efficiency and sufficiency" [42] (p. 189) may be found in the CFE.

Fourthly, exploiting old technologies in new ways provides a means to engage with new customers through new technologies and by creating new business models to disrupt markets [3]. An example would be when scientists at IBM began to track Sputnik (the first artificial Earth satellite) in space for their own amusement, which then led to the development of the technology now used for GPS [58]. This system was initially used to locate and track between our planet and space and is now used by consumers in their everyday lives [58]. Accessible through smartphone technology, from finding locations and tracking their loved ones to checking into social media and being globally connected, smartphones have centralised technology within everyday lives [58]. The development of smartphone applications encourages consumers to share their lives online and offers personalised marketing [8]. This technology could support the development of the CFE. Historically, consumers shared garments within informal networks of hand-me-down clothes in families and communities. New business models could be developed and supported by digital technology to widen the geographies of networks for sharing. Expanding on social media as a site for practical sharing, it could also provide visual virtue signalling repositioning social and emotional capital that will shift social norms of fashion acquisition, challenging the DSP.

Lastly, Christensen et al. [53] assert that to be disruptive, innovation should emerge from the lower end of the market and appeal to consumers through innovative attributes. Therefore, brands such as Tesla, maker of electric automobiles, are not authentically innovative, as they appeal to the high end of the market and therefore are not affordable and accessible to a new market of consumers; rather than disrupting business models and technology, Tesla sustains innovation within the automobile sector. The lower end of the market represents consumers least willing to pay for the product [51], as represented in Figure 4, which fast fashion is currently sustaining within the DSP. Therefore, innovation that disrupts this DSP will need to entice low-end consumers by creating new values at a similar price point to move into the mainstream. Collectively, the proposed model of Figure 4 presents Christensen et al.'s [53] elements as imbued within social and emotive contexts, combining the conceptual and theoretical frameworks to demonstrate how to disrupt the DSP with the CFE. The next section introduces the case study and will examine how it relates to Figure 4.

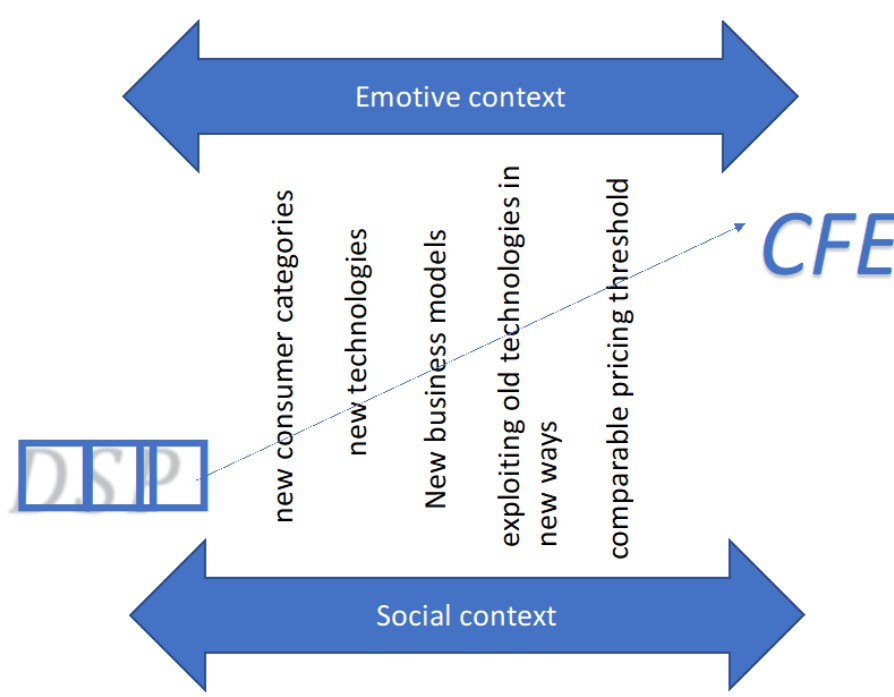

**Figure 4.** Intersections of the conceptual and theoretical frameworks with the DSP.

## 4. Case Study: Examination of an Online Fashion Rental Platform: Wardrobe as a Circular Business Model

Creswell (2007) recommends a case study approach to an inquiry when exploring how culture operates within a bounded system. Indeed, the seminal work of Christensen [17] was based on a set of case studies. As the purpose of this paper is to examine a CFE business model that competes with fast fashion, a case study provides a methodology for closer investigation of a specific issue in a real-life setting [59]. The case under examination is Wardrobe (https://www.joinwardrobe.com), a peer-to-peer fashion-renting platform offering access to borrowing celebrity and influencer wardrobes. Based in New York, Wardrobe was founded by American entrepreneur, philanthropist, and artist Adarsh Alphons in 2018. Although Wardrobe positions itself as "the living archive of fashion sitting at the intersection of sustainability and influence" [60], the original premise of the business inception was to maximise real estate space by liquidising wardrobe content rather than addressing sustainability or creating a CFE business. For the first three months of operations, prior to the COVID-19 pandemic, garments were collected from and dropped off at drycleaners, who ensured that the garments were clean and in good condition. During the pandemic, exchanges moved online, with delivery and collections from couriers via the drycleaners. Over the last few years, Wardrobe's operations have grown tenfold, demonstrating the appeal of this business model in surviving the pandemic disruption during their infancy of three months of trading. In conversation with the marketing manager, Nina Rowan, growth was sustained during the pandemic, despite the social restrictions, as young people rented fashion to generate content for their social media. This demonstrates the value of social capital created through social media (Figure 2) via engagement from Generation Z and suggests that, despite social confinement leading to social media as being their sole external window, engaging with Wardrobe provided hedonistic emotions that alleviated the monotony. Wardrobe enables access to luxury branded fashion such as Louis Vuitton, Prada, Celine, and Chanel, and for around half of the borrowers, it is their first experience with luxury fashion. Borrowers can also rent iconic vintage and celebrity wardrobes, which includes fashion worn in films, TV, and music videos and on the red carpet. For example, Queer Eye star Antoni Porowski's closet collection is for rent, including the shirt he wore in a Taylor Swift video and the black leather jacket he wore on Queer Eye that was immor-

talised in Porowski's Lego figure. Hence, the unique history of the product adds value to luxury brands that cannot be found elsewhere.

Having acquired consent from the CEO to gather data, this paper reports on Wardrobe's operations over 2021 and 2022, when the average price of a four-day rental was USD33.10; since Wardrobe was an intermediary of the renters' wardrobe/closets rather than investing in inventory, the price to rent garments was comparable to that of average fast fashion. Engagement navigates between inviting celebrities and influencers to rent their wardrobe on the platform and the renting process. Renters can maximise their inventory by collecting revenue on their assets within 2 h of posting, and the highest earning lender has, to date, earned USD10K+, which is symbolic of the micro-entrepreneurship noted by McFarlane et al. [36]. There has been less focus on marketing, with the priority on streamlining efficiencies in the users' experience (both those who rent out their wardrobe and those who rent the garments) and using technology to maximise the turnaround of cleaning and shipping. Marketing is user created [8], where renters post their activities on social media and use #Wardrobe, with the company then sharing those stories on social media.

Although the research could have adopted interviews or surveys with the users of Wardrobe or the influencers/celebrities who rent their wardrobes on the platform, it was decided that a case study would be more appropriate to analyse the business model as the focal operation of innovative commerce, as was argued by Arrigo [3]. Extant sustainable fashion literature has found that fast fashion and sustainable fashion are incomparable in terms of price, accessibility, and style from a myriad of methodological approaches across the positivistic and interpretive spectrum, leading to a saturation of understanding how to advance the sustainable fashion agenda with no clear way forward in advancing the CFE. Yet, Wardrobe does not compete on sustainability credentials; rather, it is the social and emotive context that underpins this business model, and this is a novel approach that can be replicated to advance the CFE. Wardrobe also demonstrates that social media plays a pivotal role in co-creating identity narratives between Wardrobe, influencers/celebrities, and those renting the fashion. It is these identity constructions that are of interest in driving new social norms of fashion practices and re-socialising consumers away from the DSP. To our knowledge, the application of marketing tactics to encourage engagement in the CFE has not been examined previously. Further, viewing the CFE through the theoretical framework of DI is also unique, as previously, sustainable fashion and the CE were considered sustaining innovations by competing through sustainable production and appealing to ethical values. Over the last decade, sustainable fashion has struggled to compete with fast fashion, establishing the case for disruptive tactics. This approach aligns with Goffin et al.'s [18] assertion that a case study is useful for exploratory research as well as "theory-building, theory-elaboration and theory-testing" (p. 595), which begins with explaining how the case was selected for theoretical reasons.

Following Stake's (cited in [59]) intrinsic case study procedures, the authors recognised that Wardrobe challenged the DSP of the fast fashion industry with a strategy that could be disruptive, thus adopting what Creswell terms "purposeful maximal sampling" (p. 75). A singular focused in-depth case study offers strategic insight into an issue, and although this does not enable generalisation, it offers an undiluted insight [59]. Adopting Yin's [61] embedded analysis to focus more strategically, the conceptual framework presented above was inductively developed from extant literature in order to understand the context in which fashion operates within society and which factors hinder the progression of a CFE. The case study data (see Table 1) were deductively considered against the conceptual framework to enable a pathway for disrupting the DSP.

**Table 1.** Case study parameters.

| Case | Wardrobe |
| --- | --- |
| Data sources | <ul><li>Website</li><li>Interview with the founder</li><li>Asynchronous interviews with the marketing manager</li><li>Presentations from the marketing manager</li><li>Social media; #Wardrobe</li></ul> |

Goffin et al. [18] advocate that the quality of case study research depends on the development of the theory and the appropriateness of the case to be studied. As researchers with a background in fashion marketing, retailing, social media, and consumer behaviour, specifically through the lens of sustainability, our observations and familiarisation with the relevant literature over the last decade, along with our contributions to the literature, have enabled an insight into the challenges of advancing the sustainable fashion agenda. This has led to our conceptual framework presented above, where we identified a case (Wardrobe) that offered an appropriate lens to examine the research problem [59] through the conceptual framework (theory building) and testing the theory of DI (theory elaboration) [18]. Following the advice that Goffin et al. [18] offer of evaluating the application of case study research in the innovation literature, the details of the research design, data collection, and data analysis provide transparency in the processes. In their study, Goffin et al. [18] did not identify any variation in the quality of papers that were based on single, two, or three cases; therefore, in our paper, we offer no comparison to Wardrobe to distract from the focus of this unique business model. As a caveat, we could not source another fashion-renting platform that enabled the ability to borrow influencers'/celebrities' own clothing. Therefore, the value of this paper lies in the conceptual and theoretical development of transferring the approach of the DSP to the CFE in such a way that centralises social and emotional capital by "theory-testing" the innovative model of the case study of Wardrobe.

As Table 1 displays, data did not emerge from one source, and it is pivotal to observe Wardrobe from the consumer perspective, because despite increased awareness of the impact that fast fashion has on the environment and allegations of exploiting fashion workers, as noted above, consumers continue to purchase fast fashion and apply a myriad of excuses for doing so. Therefore, the website was the first unit for analysis in May 2022. Secondly, to better understand how the business model was designed, an online interview with the founder and CEO was carried out in June 2022, adopting a semi-structured format with questions inspired from the literature. Thirdly, as marketing is often blamed for encouraging frequent impulse consumption using manipulative tactics, we were interested in how the marketing manager approached designing content and campaigns, which was explored in asynchronous interviews. Interestingly, the marketing followed two strategies: encouraging influencers/celebrities to rent out their wardrobes through the platform and encouraging consumers to borrow influencers'/celebrities' clothing. Therefore, we also had access to the content that was presented to the influencers/celebrities as a business proposal and observed the co-created marketing on social media. Finally, we reviewed the social media network Instagram for Wardrobe to observe "instances" of content using #Wardrobe. These parameters are similar to the main data collection materials from the case studies examined by Goffin et al. [18].

*4.1. Data Collection*

The marketing manager of Wardrobe contacted one of the authors in November 2021 after reading their research, and this began an exchange of discussions through email and online meetings, during which the marketing manager presented corporate information that aided the understanding of the business model operations from inception. In between those discussions, observations were made of the website content and the marketing campaigns of new influencers and celebrities who were sharing their Wardrobes to the renting client base

were received. Collectively, the importance of capitalising on social and emotive capital—the opportunity for consumers to acquire status fashion to present themselves in their social worlds and in which they could be seen—became apparent; this has been missing from previous literature examining fashion consumers' perspectives of sustainable fashion. It is also important to note that communications began in November 2021, with the previous two years having been disrupted by the global COVID-19 pandemic. Fashion retailers had struggled due to the forced closure on non-essential retailing, and the restrictions had reduced the kind of social occasions that often trigger fashion consumption [13]. In contrast, Wardrobe reported tenfold growth during the pandemic years, as Generations Z and Y borrowed celebrity luxury fashion to post selfies on social media wearing the influencer/celebrity outfits in their homes. The novelty of this was amusing, but more importantly, the practice aligned with social and emotive capital that avoided the DSP; rather, this practice endorsed the CFE and highlighted Wardrobe as a disruptive and innovative business model using similar DSP marketing tactics, and as such, it merited further investigation to map out the conceptual and theoretical implications to aid in understanding of how to advance the CFE.

Analysis included an ethnographic evaluation of the sample in its setting [59], as presented in Table 1 above. However, the conceptual and theoretical frameworks also inform the setting—the broader societal and commercial constructs within which consumers operate and thus support theoretical development of how the CFE can disrupt the fast fashion paradigm. The frameworks offer a strategic approach to collecting relevant instances that exemplify meaning; therefore, the analysis is deductive as being informed by the conceptual and theoretical frameworks, with themes that were extracted from the literature reviewed. As our previous research has explored facets of fashion consumption and sustainable behaviours, it was recognised that the business model of Wardrobe addressed the social and emotive contexts often missing with the CFE. Therefore, the conceptual and theoretical foundations were examined with the data parameters presented above. Having established the conceptual framework and identified an appropriate theory, the analysis of the case study was independently carried out by both authors to support inter-coder reliability [18]. Table 2 presents a summary of the data collection, timing, and analysis.

**Table 2.** Data collection and analysis.

| Data Sources | Collection of "Instances"—Examples | Interpretation |
|---|---|---|
| Website Timing: May 2022 | Information—consumer value with a focus on:<br><br>• Celebrity/influencer lifestyle—unique accessibility<br>• Accessibility to a wide range of luxury brands<br>• Rental = low cost<br>• Sustainability<br>• Income generation<br><br>Summary of website positioning:Fast fashion attributes of a wide range of luxury brands, low-cost, and sustainable = strong appeal to Gen Z in developing social identity and status. Development of CFE—appeals to social/emotive context and social capital | Disruption: new business model—NEP |
| Founder Timing: 23 June 2022 | Identification of problem—storage within real estate<br>Additional storage not the answer<br>Wider implication of limited storage capacity and impact on the environment<br>Spark of solution—developing business model—renting using celebrities, providing unique value<br>"Sharing economy platform"<br>Development of CFE<br>Disruption with a focus on unique value, renting, income, and sustainability. | Disruption: new business model—new P2P rental model—NEP |

**Table 2.** *Cont.*

| Data Sources | Collection of "Instances"—Examples | Interpretation |
|---|---|---|
| Marketing Manager Timing: 17 and 18 November 2021 | Developing CFE platforms Consumer use of influencers—entertainment—renting—Gen Z social media—social capital Sustainability Communication strategy—social media and hashtag strategy—emotive context Income generation—time context—emerging from COVID-19 | Disruption: new consumers to rental—influencers/consumers |
| Presentation from Marketing Manager Timing: August 2021 | Focus on lifestyle value of celebrities/influencers Social capital Accessibility to unique fashion Rental model P2P—income generation Sustainability "The living archive of fashion sitting at the intersection of sustainability and influence" "Democratising fashion" "Wardrobe liberation" | Disruption: exploring technologies in new ways new consumers to rental market |
| Social Media—Instagram: #Wardrobe Timing: November 2021 | Wardrobe hashtag strategy Influencers—lifestyle, e.g., Oliviaculpo@Instagram—4.9M followers Gen Z—significance of emotive context and social capital Proceeds of income donated to charitable causes—Endofound.org Renting and income generation Sustainability | Disruption: exploiting existing technologies with the role of social media and influencers |

Having established the functionalities of the case study and the rationale for selecting Wardrobe, the paper will examine the implications of Wardrobe as a CFE example and how this responds to the conceptual and theoretical framework presented in Figure 4 above.

*4.2. Wardrobe Opens up New Consumer Categories*

New consumer categories are evident in both the influencers/celebrities who rent out their wardrobes and the borrowers who may be renting fashion for the first time, given that the practice is still to infuse into societal norms [2,50]. Whereas celebrities/influencers are often used as marketing intermediaries to sell the inventory of fashion brands [21,33,36], Wardrobe is unique in being the intermediary between celebrities/influencers and consumers, who are often admirers of the celebrity. Marketing that depicts fashion-focused lifestyles deepens the emotive connection between the celebrity and the audience. As Alphons recognised, the relationship moves beyond the digital realm to "wear[ing] that jacket in real life, it's kind of mind blowing in that way [that's just] not possible in any part of the world, apart from the world we've created." Utilising this social capital not only provides a unique experience but can also generate excitement that is co-created between Wardrobe, the influencer/celebrity, and the consumers renting the fashion, especially through visual presentations on social media where hashtags (#) can be used to connect all co-created stakeholders and provide a visible external indication of an aspirational "good life" [15,25,28–30].

New consumer categories are also formed, as 50% of Wardrobe's Generation-Z consumers experienced luxury branded and iconic vintage fashion for the first time, as the pricing is affordable for the average consumer [2,10]. Therefore, Wardrobe creates disruption from the lower end of the market [53], where consumers' pricing thresholds are reduced [23]. The opportunity to rent luxury fashion may supersede concerns around ownership found in previous research [42], especially if Generation Z prefer to not repeatedly wear the same garments [14,22], as this is a sector from which many fashion consumers are locked out due to higher pricing [23]. Consequently, Wardrobe enables access to a variety of luxury and celebrity fashion, which can be used to replicate aspiration and success [15]. The role of highly desirable luxury brands becomes a prop within the construction of

self-identity to project a desired image socially [26], creating aspirational social capital on social media. This blurring of entertainment and commercial activities on social media platforms [8,21] has long been a tactic of fashion marketing [4]. Yet, within this practice there is also the opportunity to renegotiate the stigma that has historically been found in the sharing of garments, especially in hand-me-down clothes that were passed between families and communities [23]. As such, this sharing model emerges from the luxury sector and celebrity/influencer capital [21,34], supported by digital technology to present this desirable hedonistic experience socially. The model supports widening geographies of networks for sharing and could potentially reduce preferences for ownership, especially when luxury and vintage fashion can be more playful and experiential. This has been enhanced by new technologies.

### 4.3. New Technologies

Wardrobe benefits from new technologies that have made renting fashion easier to engage in [3], as well as connecting influencers/celebrities with their fan base [31]. This is increasingly utilised by online fast fashion marketers to entice frequent impulsive fashion consumption, creating a saturated and highly competitive industry where advantages are found in speed to market and lower price points [22]. However, this comes at a cost: exploitation of people and the planet [6]. As consumers are increasingly aware of the impact of fast fashion on the climate crisis and have expressed discomfort with the way in which fast fashion employees are treated [14,19,24], renting becomes a viable alternative that enables the utilitarian maximisation of scarce resources from the cradle to the grave [40]. Although Wardrobe satisfies those least willing to pay [51], it also aligns with status and esteem to develop a new purpose of renting, which addresses both the social and emotional context sought by Generation Z [14,19,22]. This has the potential to reshape established social norms of fashion acquisition and provide a USP of sustainable advantage [55]. To make this more appealing to consumers, Alphons explained how they track data to understand and improve the user experience for both influencers/celebrities who lend their fashion and consumers who borrow the fashion. This includes timelines, convenience, cleaning, marketing, and browsing. Therefore, in the same way in which fast fashion has examined the user experience and offered discounts, free delivery, and payment options to ease the barriers to and risks of consumption [7,13,39], Wardrobe focus on refining this new business model to entice new consumers through new technologies, which reduce the risks and enhance the experience of acquiring fashion through the CFE.

### 4.4. New Business Models

The new business model created by Wardrobe is similar to that of Alibaba, Uber, Airbnb, and Facebook, who do not invest in inventory and generate economic capital by acting as an intermediary between owners and borrowers. This model replicates fast fashion characteristics of low prices, access to a wide range of fashion, and quick turnaround of fashion items provided by influencers/celebrities by enabling consumers to change wardrobes more frequently [10] but does so in such a way that it elevates social and emotional capital. Access to a wide range of luxury fashion merchandise and high rental turnover at a low price resonates with Generation Z's need for fashion acquisition [13,22], which is still driven by style, price, accessibility, and branding [14,15]. Thus, "Sufficiency" [42] (p. 189) is evident, as Wardrobe still resembles the fashion hauls depicting variety, but this is underpinned by a circular model that avoids landfill contributions [22,43]. The model is augmented by enhancing the user experience through technology to make the process more efficient, user friendly, and sustainable. In addition to recognising the role of social and emotive context, this business model offers alignment with fashion self-identity and sustainability: no sacrifice required. This is an illustration of how Wardrobe is becoming a destination for Generation Z and fashion creators.

Although the delivery of the garments is not instantaneous, Wardrobe is as accessible as buying fast fashion online, with a unique USP of belonging to a celebrity/influencer and

social capital that supersedes the rapid style change characterised by fast fashion. Similar technologies that enhanced the fast fashion consumer experience by minimising risk and effort for consumers [13] can align with the sustainability agenda, as we cannot buy our way out of the climate emergency [41]. The role of celebrities/influencers is paramount to the success of this business model. As celebrities/influencers are renting their own wardrobes, they are more authentic, especially as there is no sponsored content with brands or advertisers. Authenticity helps to develop a sense of trust with their followers and contributes to the success of the business model. Financial benefits for the celebrities/influencers are achieved through rental, and this platform provides celebrities/influencers an opportunity to communicate their wider values. For example, rental proceeds can be donated to a charitable cause. As the business model moves beyond any one brand and operates on celebrity capital to generate social capital, Wardrobe is exploiting the social capital of fast fashion marketing to direct consumers to the CFE.

*4.5. Exploiting Old Technologies in New Ways*

Exploiting old technologies in new ways is less demonstrable in a technological sense, as the amalgamation of fashion, identity, and social media is not a new concept. It is ingrained within fashion discourse, and fashion is dependent on gaining social acceptance to gain traction [8]. Similarly, social acceptance is crucial to advancing rental fashion [2], which illuminates the opportunity to mimic fast fashion marketing and permeate societal norms. Fashion consumers look to social media for fashion inspiration [24], and Wardrobe provides a similar online experience that provides the lifestyle experiences that Generation Z crave [4]. Content is stimulated by with #drops and new content that satisfies the hedonism of impulsive and frequent fashion acquisition [13]. although social media enables branding to be co-constructed between the brand's communications and consumer interactions, Wardrobe focus on sharing user experiences as their marketing tool, maximising the word-of-mouth exchange [24] and embedding authenticity directly from the influencer/celebrity to the consumer. The benefits of this focus on the user experience to encourage more consumers to rent from the CFE and make the experience less risky [13]. Communicative exchanges through social media increase visibility [44] and mimic the blurring of fashion entertainment and commerce in which consumers have been socialised [9,10]. The model does not require Generation Z to give up any of their values in fashion consumption; rather, it allows them to be enhanced and is responsive to alleviate the guilt experienced from fast fashion consumption [14,23]. Generation Z can engage with experimenting with fashion styles and constructing fashion self-identity through continued engagement with Wardrobe. Collectively, this suggests that renting platforms have the potential to disrupt the DSP.

The Wardrobe case study is an illustration of disruptive innovation that can progress the CFE. Despite the model emerging from an economic perspective, it addresses sustainability by minimising the excavation of scarce natural resources for limited use before being disposed in landfills [4,22,43]. Considering the rapid growth of the business in a short time frame, harnessing technology to enhance the service attributes of renting could potentially pave the way for the company to become a market leader. The disruptive and innovative model that exploits the current DSP ensures that Generation Z do not need to sacrifice social and emotional capital value in their fashion practice; if anything, this model enhances their overall self-identity and provides egotistical value, as the model allows them to take account of their sustainable contribution. Additionally, if the rental is delivered by courier, the same way that fast fashion is delivered, then consumers will still experience the hedonistic thrill of receiving a new (to them) garment to wear and be seen in, as well as experiencing the thrill of unpacking [13,48].

To map out the ways in which Wardrobe disrupts and innovates the DSP of fast fashion, Table 3 captures the main points from the conceptual framework that have been presented above through the lens of DI. In the first column, the mechanisms that underpin fast fashion management and marketing are established as a basis of comparison, fol-

lowed by the components of Wardrobe and how this creates appealing values. Having demonstrated Wardrobe as a CFE business model that can provide added value for fashion consumers, concluding comments, limitations, and recommendations for future research are provided next.

**Table 3.** Comparative analysis of the intersections between the conceptual and theoretical frameworks of the DSP and CFE as experienced by Generation Z.

| Conceptual Categories | Fast Fashion (DSP) | Wardrobe (CFE) | Appealing Values |
|---|---|---|---|
| Business model | B2C Transaction | P2P rental | From ownership to rental |
| Product offering | Wide assortment | Mimic wide assortment | Mimic fashion Construction |
| Product positioning | Instant access to fashion trends | Instant access to luxury fashion | Mimic instant access to hedonic lifestyle |
| Product turnover | Rapid fashion turnover | Mimic rapid rental turnover | Mimic rapid styling |
| Product experience | Ownership | Rental | Mimic fashion construction and sustainable experience |
| Price | Low cost | Mimic low cost | Low cost |
| Promotion—digital | Social media brand community with or without influencer collaboration Brand | Social media—Wardrobe platform—celebrities/influencers Wardrobe website | Mimic Social capital Emotional capital "The good life" Social media |
| Place | Brand website | P2P platform | Mimic delivery: hedonistic experiences |
| Business model Implications | Revenue through transactions | Revenue through rental platforms | From a transactional to a circular sharing economy |
| Sustainability | Environmental costs | Environmental gains | From disposal to reuse |

## 5. Concluding Comments

That the CFE has not yet disrupted the incumbency of the DSP is not an indication that it has failed to do so. As Schmidt and Druehl [51] assert, disruption often takes time to encroach on market share and embed new consumer practice, and it may never fully displace fast fashion. However, Jain et al. [2] considered that future research should focus on understanding consumer needs, motivations, and preferences for renting fashion and provide strategies for marketing managers; however, this does not include using the same strategies as fast fashion marketing to appeal to the same fast fashion consumers. Examining the business model of Wardrobe has illustrated that there is potential to for the CFE to disrupt the DSP of fast fashion using a similar approach. Further, Jain et al. [2] suggest that marketers of fashion rental businesses could promote the utilitarian and hedonistic values that encapsulate the social and emotive capital of being able to access a large quantity of luxury fashion, and our research has identified the benefits of doing so. Value can be co-created between the business, influencers/celebrities, and consumers on social media. Indeed, this seems to supersede social activities to become a way to present oneself on social media to curate social capital. Although Jain et al. [2] also suggest that the environmental benefits also be included in marketing, extant research has already shown that this alone will not change behaviours; nevertheless, as a secondary consideration, it will lessen the guilt and cognitive dissonance of fashion acquisition and offer a tenant of value [4,5].

This novel paper makes four theoretical contributions to the literature. Firstly, the paper is one of the first to argue that development of the CFE is dependent on recognising the importance of addressing social and emotional capital to engage fashion consumers in adapting their fashion practice. Although the sustainability aspect is important, it is a secondary value that appeals to consumers, albeit it will endorse the practices of

the CFE, and this will become increasingly important as concerns grow for the climate emergency [24]. Therefore, combining the conceptual model developed from reviewing the literature with the theory of DI advances both sustainable fashion and innovation literature by illustrating that embodied social capital should be prioritised in the inception and marketing of the CFE. A second contribution is the unique examination of fashion, the conceptual model, and the CFE through the lens of a DI theoretical framework. This has enabled a deeper understanding of how the fast fashion model has dominated the marketplace and opens the debate that similar tactics can be utilised to entice consumers to the CFE. This offers a paradigmatic conceptualisation of how to market the CFE through developing innovative business models and addresses the attitude–behaviour gap that has existed over the last two decades [16]. The case study of Wardrobe exemplifies that engaging with the CFE does not require a sacrifice of social and emotional capital—rather, it can enhance the experience with egotistical value. Thirdly, examining the CFE through the lens of DI indicates that many of the disruptive models that have evolved seamlessly into society and penetrated everyday lives originally evolved through other agendas—such as Sputnik leading the development of the technology now used for GPS ([58], as described above). Similarly, Wardrobe emerged from the desire to maximise economic utility and, through this, identified a business model that could disrupt impulsive and frequent fast fashion consumption while also addressing sustainability. Fourthly, the argument that the DSP of fast fashion can be disrupted by using similar management and marketing tactics is a novel idealisation. Previous research has considered fast fashion an oxymoron to sustainability, and as consumption was stimulated by marketing, the link between mimicking fast fashion marketing to encourage CFE acquisition had not been recognised. However, given that consumers are reluctant to sacrifice their social and emotional capital, despite increasing concern for the climate emergency, this unique innovative approach may be pivotal in growing sustainable fashion practices.

*Limitations and Future Research*

This paper has focused on one case as an example of how the CFE can disrupt the fast fashion industry, and future research could expand to multiple cases for comparison and to illuminate different perspectives [59]. The approach outlined in this paper is novel, emerging from understanding the sustainable fashion and fashion marketing literature over the last few decades. Although previous research has examined consumer experiences and perceptions of sustainable fashion and considered the sustainability of new business models, to our knowledge, a CFE example has not previously been explored within the conceptional and theoretical frameworks as presented above. Yet, there is much to learn from the application, despite the narrow perspective. As such, this paper does not provide a generalisation of the CFE; rather, it is an example of related concepts examined against the theory of disruptive innovation. It is also recognised that the application of DI is contentious (cf [62]); however, the debate around disruption provides a framework for a deeper analysis and is not one that usually examines the fashion industry.

The fashion industry has always been reflective of looking back and looking forward for design and material inspiration [4]. Many fashion actors have been described as disruptive—for example, Vivienne Westwood and Alexander McQueen—and been applauded for innovating ideas supported by technological advances. Christensen et al. [53] responded to critiques of the theory of DI, citing the numerous dynamic variables that change the way in which the theory can be applied, and welcomed further study. Our paper contributes to this debate and further challenges the fashion industry to respond to the wicked problem of our time, the climate crisis, and by analysing DI alongside extant literature to identify solutions. Therefore, our paper emerges from immersion in the fashion, marketing, and sustainability literature and invites contributions to furthering this conversation. Other opportunities for deepening understanding of the CFE include exploring the consumer experience of the sharing economy, as concluded by Jain et al. [2], which

could be supported by adopting netnography to illuminate this social media window as representing social capital or immersive qualitative methods to reflect the lived experience.

One last aspect of a limitation and how it can be addressed in future research is the reliance on moving garments around using couriers, which creates other sustainability issues and carbon emissions [2]. If consumers are expressing concern for sustainability, it will be an important issue to address and it will be important to provide transparency; yet, this can also be used as a marketing tool. For example, the Scottish rental company Advanced Clothing Solutions (ACS) offer high street fashion retailers a repair and resell service for returned clothing [63]. ACS (2023) support the expansion of the CFE, and amongst their many accreditations they are both B Corp Certified and carbon neutral. As such, it is an organisation that can be integrated as a marketing tool to illuminate the efforts made to reduce the carbon footprint of renting fashion, including cleaning, repairing, and transport. Although we can recognise that Wardrobe protects scarce natural resources, maximises the lifespan of garments, and minimises what is sent to landfills, it is also acknowledged that this is part of a solution, and more work is required to implement sustainable principles in all operational aspects. Illuminating those processes aids the issue of transparency.

**Author Contributions:** Conceptualization, E.L.R. and N.S.; Methodology, E.L.R. and N.S.; Formal analysis, E.L.R.; Investigation, E.L.R.; Resources, E.L.R.; Data curation, E.L.R.; Writing—original draft, E.L.R.; Writing—review & editing, E.L.R. and N.S.; Visualization, E.L.R.; Supervision, E.L.R. All authors have read and agreed to the published version of the manuscript.

**Funding:** This research received no external funding.

**Institutional Review Board Statement:** The study was conducted in accordance with the Declaration of Helsinki, and approved by the Ethics Committee of Glasgow Caledonian University on 1 June 2022.

**Informed Consent Statement:** Informed consent was obtained from all subjects involved in the study.

**Data Availability Statement:** Data is not available.

**Conflicts of Interest:** The authors declare no conflict of interest.

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
