# Peer review of "Fashioning the Circular Economy with Disruptive Marketing Tactics Mimicking Fast Fashion’s Exploitation of Social Capital: A Case Study Exploring the Innovative Fashion Rental Business Model “Wardrobe”"

_sustainability, doi:10.3390/su151914532_

Round 1
Reviewer 1 Report
Dear Authors,
Thank you for your work. I really appreciate the topic; it is an exploration of an marketing and sustainability area and it's really interesting. I think that many other works could follow this work in the field. The consideration of the social and emotional side of the topic and the theories behind are well explained in the introduction and the second part of the text. the methodology is surely well fitted to the explorative context as sustainable fashion is still niche and new. I should also add that the disruptive innovation and the approach of Generation Z are important for the work, and they are both investigated in the paper. I really liked figure 1. Case study is ok! and is a good way to explore the area via "Wardrobe". The conclusion and limitations are also good. I liked the paper and hope the other reviewers and readers will also like it.
Best Regards,
Author Response
Thank you do much for your positive review, it is great to find a like mind for our conceptualisation of this idea, which we feel is quite different. I do hope it inspires similar approaches to research and we felt it was important to capture the social and emotional side of fashion consumption. Your supporting comments are much appreciated.
Reviewer 2 Report
It would be necessary to have the interviews.
Author Response
Thank you for your feedback, we have provided a table with more information on the interviews along with the other data collection to demonstrate how this collectively aligns with the case study.
Reviewer 3 Report
The author presented an interesting article on fast fashion but not clearly presented. The framework lacks the literature support. The authors mentions about data collection but missed to present the analysis.
The writing style requires improvement.
Author Response
Thank you for your feedback on our paper. We understand that you found it interesting, but that it was not clearly presented. We wondered if you could elaborate on what was not clear in the paper? You also mentioned that the framework lacks the literature support, however we have covered the literature that helped develop the conceptual and theoretical framework, and wondered if you could direct us to the missing literature? We also note that we missed to present the analysis from the data collected, however this was a case study and as such reported on how the business model can be viewed through our conceptual and theoretical framework. We welcome further information to support this review, but as it stands felt unable to progress any amendments based on the feedback.
Reviewer 4 Report
The authors argued that there are lessons to be learned from the tactics of the fast fashion industry that can be used to enhance CFE. The research topic is focused on current content and has unique perspectives.
Abstract: line 28. The authors mentioned, “There has been limited research examining the CE and CFE.” However, there are so many research articles focusing on CFE. I remembered a special issue featuring CFE in this journal a few years ago.
P.2: line 88~91. Although the study focuses on the CFE business "Wardrobe," the authors did not explain what a "Wardrobe" is in the Introduction. Please provide a brief explanation of it. Additionally, I understood the benefits and drawbacks of the fast fashion business model. However, since the main focus is on CFE, the authors should mention the benefits and drawbacks of CFE or the rental wardrobe business model.
P.2: line 91. The authors should briefly explain why Christensen's disruptive innovation is used as a theoretical framework in this research.
P.3. line 101~121. The authors' conceptual model overemphasizes the impact of social media and social capital on clothing consumption by consumers in Generation Z. Your opinion implies that fast fashion companies used social media and influencers to influence Gen Z consumers to purchase fast fashion clothing, but this is only partially accurate.
P.3. Figure 1: The title of Figure 1 is “Conceptual framework: Understanding the social and emotive contexts of fast fashion.” I suggest the authors add “fast fashion disruption” before the CFE disruption.
e.g.
Social context → emotive context → fast fashion disruption → CFE disruption
P.9 line 360~363. The authors noted, “While fast-fashion retailers harness technology to augment the consumer experience, through minimizing risk and effort for consumers and offering personalized marketing, this does not support the sustainability agenda in addressing the issues of concern to Generation Z.” Sustainable fashion companies are also using personalized marketing because I have received many targeted advertisements and emails from them. How can the authors claim that sustainable fashion companies do not use technology for personalized marketing?
P.17. Line 730~732. In concluding comments, the authors noted, "The paper is one of the first to argue that development of the CFE depends on recognizing the importance of addressing social and emotional capital to engage fashion consumers in adapting their fashion practice.” Line 736~739. “Therefore, combining the conceptual model developed from reviewing the literature with the theory of DI advances both the sustainable fashion and innovation literature by illustrating that embodied social capital should be prioritized in the inception and marketing of the CFE.”
The conclusions imply the fast fashion strategy utilizes social capital between influencers and their audiences to boost clothing sales. Therefore, the authors posit utilizing a fast fashion company's strategy is practical to apply for CFE. However, I have a differing opinion on the strategy of CFE company presented by the author. I would like to know whether using social relationships to promote purchasing clothing aligns with sustainability. What are the author’s thoughts on whether this strategy is viable for sustainable fashion? How do the authors define sustainable fashion or circular fashion economy?
Author Response
Thank you so much for taking the time to review our paper, your consideration is very much appreciated. We have responded to each of your comments below.
The authors argued that there are lessons to be learned from the tactics of the fast fashion industry that can be used to enhance CFE. The research topic is focused on current content and has unique perspectives.
Thank you for recognising this novel approach. We very much enjoyed responding to your thoughtful review.
Abstract: line 28. The authors mentioned, “There has been limited research examining the CE and CFE.” However, there are so many research articles focusing on CFE. I remembered a special issue featuring CFE in this journal a few years ago.
We have revised this to reflect your comments. This sentence now states: Thirdly, while there is an emerging literature stream examining the CE, and CFE, this focuses more on consumer practice and behaviours: and little attention has been paid to how the CFE can be marketed to engage with consumers
P.2: line 88~91. Although the study focuses on the CFE business "Wardrobe," the authors did not explain what a "Wardrobe" is in the Introduction. Please provide a brief explanation of it. Additionally, I understood the benefits and drawbacks of the fast fashion business model. However, since the main focus is on CFE, the authors should mention the benefits and drawbacks of CFE or the rental wardrobe business model.
This was remiss of us! And we have now rectified this, as evident below:
Therefore, this paper will present an established CFE business. Wardrobe’ as a case study. Wardorbe is an online a peer-to-peer fashion sharing platform operating in the USA that enables consumers access to luxury brands as well as celebrity wardrobes. We examine Wardrobe’s business model through a conceptual framework that was constructed from reviewing the fashion and marketing literature along, with the theoretical framework of Disruptive Innovation developed by Christensen [17].
P.2: line 91. The authors should briefly explain why Christensen's disruptive innovation is used as a theoretical framework in this research.
Again, this was remiss and we have included the following sentences:
Disruptive Innovation has been successful in determining shifts in the marketplace that are supported by technological advancement and provides an opportunity to understand market innovations that disrupt current ways of business operations. Market expansion is often sought by moving into new geographical areas, yet there can be new forms of business operations that can support growth, and this has been evident in disruptive business models in the sharing economy, such as Uber and Airbnb.
P.3. line 101~121. The authors' conceptual model overemphasizes the impact of social media and social capital on clothing consumption by consumers in Generation Z. Your opinion implies that fast fashion companies used social media and influencers to influence Gen Z consumers to purchase fast fashion clothing, but this is only partially accurate.
While this may only be partially accurate, it is still a tactic of fast-fashion marketing to stimulate consumption through social media and Gen Z is particularly susceptible to this, and it is well documented in the fashion marketing literature, and it is indeed the main way in which Wardrobe markets. Many fast-fashion retailers here in the UK use influencers as marketing – there are so many ongoing examples, the most recent would-be Naomi Campbell for PLT and anyone who has been on Love Island. There is also much evidence of this on social media. So, it is not clear what change you would like made here, for this is the thrust of the paper – social media is used for marketing fast-fashion, but could be used to market sustainable behaviours as represented by the circular economy. Any further elaboration would be appreciated as a guide for amendments.
P.3. Figure 1: The title of Figure 1 is “Conceptual framework: Understanding the social and emotive contexts of fast fashion.” I suggest the authors add “fast fashion disruption” before the CFE disruption.
e.g.
Social context → emotive context → fast fashion disruption → CFE disruption
Thank you for this interesting thought! We considered this and discussed it, but we felt that contradicted with the aim of the paper – to disrupt fast-fashion with the CFE. We see CFE as the disruptor of fast-fashion, and not that fast-fashion is disruptive, although it was in the past. We seek to provide an alternative to fast-fashion. So we have changed it to:
Social context → emotive context → CFE disruption (of fast-fashion)
We hope that this still captures that it is CFE that is the disruptor of the incumbent fast-fashion model
P.9 line 360~363. The authors noted, “While fast-fashion retailers harness technology to augment the consumer experience, through minimizing risk and effort for consumers and offering personalized marketing, this does not support the sustainability agenda in addressing the issues of concern to Generation Z.” Sustainable fashion companies are also using personalized marketing because I have received many targeted advertisements and emails from them. How can the authors claim that sustainable fashion companies do not use technology for personalized marketing?
We can see how this has been misinterpreted. We do not mean that sustainable fashion organisations do not use personalised marketing – we mean that by fast-fashion marketing augmenting the consumers experience encourages more consumption that is an antithesis to sustainability. We do not cover sustainable fashion brands or marketing in our paper, so to make our point clearer, we have changed the sentence to:
While fast-fashion retailers harness technology to augment the consumer experience, through minimising risk and effort for consumers and offering personalised marketing, the aim here is to encourage consumption which does not support the sustainability agenda in addressing the issues of concern to Generation Z.
P.17. Line 730~732. In concluding comments, the authors noted, "The paper is one of the first to argue that development of the CFE depends on recognizing the importance of addressing social and emotional capital to engage fashion consumers in adapting their fashion practice.” Line 736~739. “Therefore, combining the conceptual model developed from reviewing the literature with the theory of DI advances both the sustainable fashion and innovation literature by illustrating that embodied social capital should be prioritized in the inception and marketing of the CFE.”
The conclusions imply the fast fashion strategy utilizes social capital between influencers and their audiences to boost clothing sales. Therefore, the authors posit utilizing a fast fashion company's strategy is practical to apply for CFE. However, I have a differing opinion on the strategy of CFE company presented by the author. I would like to know whether using social relationships to promote purchasing clothing aligns with sustainability. What are the author’s thoughts on whether this strategy is viable for sustainable fashion? How do the authors define sustainable fashion or circular fashion economy?
This is a really interesting prospect but one that is beyond the purpose of this paper. On this paper we are not exploring sustainable fashion brands, this is a sharing economy business and while it addresses sustainability – it does so overtly. Sustainable fashion brands are of interest of course, but we do not believe that more consumption, no matter how sustainable it is, will solve the climate emergency, and so direct our attention to circular fashion. We do this with an example of a rental model that capitalises on social media and social capital for operations, which we believe is a unique perspective. We do not address sustainable fashion, we address the circular fashion economy which is an alternative means of sustainable fashion. Your comment would, however, be an interesting line of enquiry to pursue - would you like to write a paper with us exploring this?!
Many thanks for engaging with our paper so carefully and providing us with thoughtful and constructive feedback, this is much appreciated.
Round 2
Reviewer 3 Report
The article covers an interesting topic and has a unique perspective related to circular economy and sustainability. The relationship between social context, emotive context and CFE disruption is well explained. Adopting a case study approach to study the relationship is good. There is a spelling mistake in Figure 1 – CFE disruption. You have presented the list of data sources in table 1 summarized the findings briefly (459 – 471). Good if you can the present the analysis / findings in table format. Overall the article looks good.
Author Response
Thank you for your positive review, we are delighted that you found the paper interesting and unique. Thanks also for noting the spelling mistake! We had not considered the inclusion of a table to present the analysis and thank you for suggesting that. It makes more sense and helps to connect the data and the meaning more succinctly, that has been added on p. 580.
Reviewer 4 Report
The author chose not to rewrite the conclusion, which is slightly disappointing.
Author Response
The conclusions imply the fast fashion strategy utilizes social capital between influencers and their audiences to boost clothing sales. Therefore, the authors posit utilizing a fast fashion company's strategy is practical to apply for CFE. However, I have a differing opinion on the strategy of CFE company presented by the author. I would like to know whether using social relationships to promote purchasing clothing aligns with sustainability. What are the author’s thoughts on whether this strategy is viable for sustainable fashion? How do the authors define sustainable fashion or circular fashion economy?
I am sorry that you had wanted us to rewrite the conclusion, and I add your feedback above so I can respond to it. We are saying that fast fashion strategy utilises social capital between influencers and their audiences to boost clothing sales. We are not suggesting using social relationships to promote purchasing clothing aligns with sustainability - because any consumption that stimulates production is not sustainable. We believe we must make use of the resources already in circulation. Therefore, we are suggesting that the fast fashion strategy utilising social capital between influencers and their audiences can boost clothing sales for circular fashion. We define the circular fashion economy in lines 37 to 70 but we do not define sustainable fashion as that is not of interest to the paper. We are observing a case of how circular fashion mimics fast fashion to market their rental service to replace the consumption act. We focused our conclusion on that rather than introducing new ideas. I hope that makes sense.